# Adversarial Robustness is at Odds with Lazy Training

**Yunjuan Wang**
Department of Computer Science
Johns Hopkins University
Baltimore, MD, 21218
`ywang509@jhu.edu`

**Enayat Ullah**
Department of Computer Science
Johns Hopkins University
Baltimore, MD, 21218
`enayat@jhu.edu`

**Poorya Mianjy**
Department of Computer Science
Johns Hopkins University
Baltimore, MD, 21218
`mianjy@jhu.edu`

**Raman Arora**
Department of Computer Science
Johns Hopkins University
Baltimore, MD, 21218
`arora@cs.jhu.edu`

## Abstract

Recent works show that adversarial examples exist for random neural networks [Daniely and Shacham, 2020] and that these examples can be found using a single step of gradient ascent [Bubeck et al., 2021]. In this work, we extend this line of work to "*lazy training*" of neural networks – a dominant model in deep learning theory in which neural networks are provably efficiently learnable. We show that over-parametrized neural networks that are guaranteed to generalize well and enjoy strong computational guarantees remain vulnerable to attacks generated using a single step of gradient ascent.

## 1 Introduction

Despite the tremendous success of deep learning, recent works have demonstrated that neural networks are extremely susceptible to attacks. One such vulnerability is due to arbitrary adversarial corruption of data at the time of prediction, commonly referred to as *inference-time attacks* [Biggio et al., 2013, Szegedy et al., 2014]. Such attacks are catastrophically powerful, especially in settings where adversaries can directly access the model parameters. By adding crafted imperceptible perturbations on the natural input, such attacks are easily realized across various domains, including in computer vision where an autonomous driving car may misclassify traffic signs [Sitawarin et al., 2018], in natural language processing where automatic speech recognition misinterprets the meaning of a sentence [Schönherr et al., 2019], among many others.

The potential threat of adversarial examples has led to serious concerns regarding the security and trustworthiness of neural network-based models that are increasingly being deployed in real-world systems. It is crucial, therefore, to understand why trained neural networks classify clean data with high accuracy yet remain extraordinarily fragile due to strategically induced perturbations.

One of the earliest works to describe adversarial examples was that of Szegedy et al. [2014], motivated by the need for networks that not only generalize well but are also robust to small perturbations of its input. This was followed by a body of works that finds adversarial examples using various methods [Goodfellow et al., 2015, Papernot et al., 2016, Madry et al., 2018, Carlini and Wagner, 2017a,b]. Although lots of defense algorithms have been designed, most of them are quickly defeated by stronger attacks [Carlini and Wagner, 2017a, Carlini et al., 2019, Tramèr et al., 2020]. Much of the prior work has focused on this empirical "arms race" between adversarial defenses and attacks; yet, a deep theoretical understanding of why adversarial examples exist has been somewhat limited.

36th Conference on Neural Information Processing Systems (NeurIPS 2022).

Recently, a line of work theoretically constructs adversarial attacks for random neural networks. Inspired by the work of Shamir et al. [2019], Daniely and Shacham [2020] prove that small $\ell_2$-norm adversarial perturbations can be found by multi-step gradient ascent method for random ReLU networks with small width – each layer has vanishingly small width compared to the previous layer. Bubeck et al. [2021] generalize this result to two-layer randomly initialized networks with relatively large network width and show that a single step of gradient ascent suffices to find adversarial examples. Bartlett et al. [2021] further generalize this result to random multilayer ReLU networks. However, all of the works above focus on random neural networks and fail to explain why adversarial examples exist for neural networks trained using stochastic gradient descent. In other words, *why do neural networks that are guaranteed to generalize well remain vulnerable to adversarial attacks?*

In this paper, we make progress toward addressing that question. In particular, we show that adversarial examples can be found easily using a single step of gradient ascent for neural networks trained using first-order methods. Specifically, we focus on over-parametrized two layer ReLU networks which have been studied extensively in recent years due to their favorable computational aspects. Several works have shown that under mild over-parametrization (size of the network is polynomial in the size of the training sample), the weights of two-layer ReLU networks stay close to initialization throughout the training of the network using gradient descent-based methods [Allen-Zhu et al., 2019, Ji and Telgarsky, 2020, Arora et al., 2019]. In such a "lazy regime" of training, the networks are locally linear which allows us to give computational guarantees for minimizing the training loss (aka, optimization) as well as bound the test loss (aka, generalization). The purpose of our work is to show the deficiency of the lazy regime even though it is the dominant framework in the theory of deep learning.

Our key contributions are as follows. We investigate the robustness of over-parametrized neural networks, trained using lazy training, against inference-time attacks. Building on the ideas of Bubeck et al. [2021], we show that such trained neural networks still suffer from adversarial examples which can be found using a single step of gradient ascent.

In particular, we show that an adversarial perturbation of size $O(\frac{\|x\|}{\sqrt{d}})$ in the direction of the gradient suffices to flip the prediction of two-layer ReLU networks trained in the lazy regime – these are networks with all weights at a distance of $O(\frac{1}{\sqrt{m}})$ from their initialization, where $m$ is the width of the network. To the best of our knowledge, this is the first work which shows that networks with small generalization error are still vulnerable to adversarial attacks.

Further, we validate our theory empirically. We confirm that a perturbation of size $O(1/\sqrt{d})$ suffices to generate a strong adversarial example for a trained two-layer ReLU networks.

The rest of the paper is organized as follows. In Section 2, we introduce the preliminaries, give the problem setup, and discuss the related work. We present our main result in Section 3, and give a proof sketch in Section 4. In Section 5 we provide empirical support for our theory, and conclude with a discussion in Section 6.

## 2 Preliminaries

**Notation** Throughout the paper, we denote scalars, vectors and matrices, respectively, with lowercase italics, lowercase bold and uppercase bold Roman letters, e.g. $u$, $\mathbf{u}$, and $\mathbf{U}$. We use $[m]$ to denote the set $\{1, 2, \ldots, m\}$. We use $\|\cdot\|$ or $\|\cdot\|_2$ exchangeably for $\ell_2$-norm. Given a matrix $\mathbf{U} = [\mathbf{u}_1, \ldots, \mathbf{u}_m] \in \mathbb{R}^{d \times m}$, we define $\|\mathbf{U}\|_{2,\infty} = \max_{s \in [m]} \|\mathbf{u}_s\|$. We use $\mathcal{B}_2(\mathbf{u}, R)$ to denote the $\ell_2$ ball centered at $\mathbf{u}$ of radius $R$. For any $\mathbf{U} \in \mathbb{R}^{d \times m}$, we use $\mathcal{B}_{2,\infty}(\mathbf{U}, R) = \{\mathbf{U}' \in \mathbb{R}^{d \times m} \mid \|\mathbf{U}' - \mathbf{U}\|_{2,\infty} \leq R\}$ to denote the $\ell_{2,\infty}$ ball centered at $\mathbf{U}$ of radius $R$. For any function $f : \mathbb{R}^d \to \mathbb{R}$, $\nabla f$ denotes the gradient vector. We define the standard normal distribution as $\mathcal{N}(0, 1)$, and the standard multivariate normal distribution as $\mathcal{N}(0, \mathbf{I}_d)$. We use $\mathbb{S}^{d-1}$ to denote the unit sphere in $d$ dimensions. We use the standard O-notation ($O$ and $\Omega$).

### 2.1 Problem Setup

Let $\mathcal{X} \subseteq \mathbb{R}^d$ and $\mathcal{Y}$ denote the input space and the label space, respectively. In this paper, we focus on the binary classification setting where $\mathcal{Y} = \{-1, +1\}$. We assume that the data $(\mathbf{x}, y)$ is drawn from an unknown joint distribution $\mathcal{D}$ on $\mathcal{X} \times \mathcal{Y}$. For a function $f_\mathbf{w} : \mathcal{X} \to \mathcal{Y}$ parameterized by $\mathbf{w}$ in

some parameter space $\mathcal{W}$, the *generalization error* captures the probability that $f_\mathrm{w}$ makes a mistake on a sample drawn from $\mathcal{D}$:

$$\text{generalization error} := P_{(\mathrm{x},y)\sim\mathcal{D}}(yf_\mathrm{w}(\mathrm{x}) \leq 0).$$

For a fixed $\mathrm{x} \in \mathcal{X}$, we consider norm-bounded adversarial perturbations given by $\mathcal{B}_2(\mathrm{x}, R)$, for some fixed perturbation budget $R$. The robust loss on x is defined as

$$\ell_R(\mathrm{w}; \mathrm{x}, y) = \max_{\mathrm{x}'\in\mathcal{B}_2(\mathrm{x},R)} \mathbb{1}\left(yf_\mathrm{w}(\mathrm{x}') \leq 0\right).$$

The *robust error* then captures the probability that there exists an adversarial perturbation on samples drawn from $\mathcal{D}$ such that $f_\mathrm{w}$ makes a mistake on the perturbed point

$$L_R(\mathrm{w}; \mathcal{D}) := \mathbb{E}_{(\mathrm{x},y)\sim\mathcal{D}}[\ell_R(\mathrm{w}; \mathrm{x}, y)].$$

In this work, we focus on two-layer neural networks with ReLU activation, parameterized by a pair of weight matrices $(\mathrm{a}, \mathrm{W})$, computing the following function:

$$f(\mathrm{x}; \mathrm{a}, \mathrm{W}) := \frac{1}{\sqrt{m}} \sum_{s=1}^{m} a_s\sigma(\mathrm{w}_s^\top \mathrm{x}).$$

Here, $m$ corresponds to the number of hidden nodes, i.e., the width of the network; $\sigma(z) = \max\{0, z\}$ is the ReLU activation function, and $\mathrm{W} = [\mathrm{w}_1, \ldots, \mathrm{w}_m] \in \mathbb{R}^{d\times m}$ and $\mathrm{a} = [a_1, \ldots, a_m] \in \mathbb{R}^m$ denote the top and the bottom layer weight matrices, respectively.

In this work, we study the robustness of neural networks which are close to their initialization. In particular, we are interested in the *lazy regime*, defined as follows.

**Definition 2.1** (Lazy Regime). Initialize the top and bottom layer weights as $a_s \sim \mathrm{unif}(\{-1, +1\})$ and $\mathrm{w}_{s,0} \sim \mathcal{N}(0, \mathrm{I}_d), \forall s \in [m]$. Let $\mathrm{W}_0 = [\mathrm{w}_{s,1}, \ldots, \mathrm{w}_{s,m}] \in \mathbb{R}^{d\times m}$. The *lazy regime* is the set of all networks parametrized by $(\mathrm{a}, \mathrm{W})$, such that $\mathrm{W} \in \mathcal{B}_{2,\infty}(\mathrm{W}_0, C_0/\sqrt{m})$, for some constant $C_0$.

## 2.2 Related Work

**Adversarial Examples.** The field of adversarially robust machine learning has received significant attention starting with the seminal work of Szegedy et al. [2014]. Most existing works focus on proposing methods to generate adversarial examples, for example, the fast gradient sign method (FGSM) [Goodfellow et al., 2015], the projected gradient descent (PGD) method [Madry et al., 2018], the Carlini & Wagner attack [Carlini and Wagner, 2017b], to name a few.

Most related to our work are those on proving the existence of adversarial examples on random neural networks. This line of work initiated with the work of Shamir et al. [2019], where authors propose an algorithm to generate bounded $\ell_0$-norm adversarial perturbation with guarantees for arbitrary deep networks. Shortly afterwards, Daniely and Shacham [2020] showed that multi-step gradient descent can find adversarial examples for random ReLU networks with small width (i.e. $m = o(d)$). More recently, work of Bubeck et al. [2021] shows that a single gradient ascent update finds adversarial examples for sufficiently wide randomly initialized ReLU networks. This was later extended to multi-layer networks with the work of Bartlett et al. [2021]. A very recent work of Vardi et al. [2022] shows that gradient flow induces a bias towards non-robust networks. Our focus and techniques here are different – we show that independent of the choice of the training algorithm, all neural networks trained in the lazy regime remain vulnerable to adversarial attacks.

**Over-parametrized Neural Networks.** Our investigation into the existence of adversarial examples in the lazy regime is motivated by recent advances in deep learning theory. In particular, a series of recent works establish generalization error bounds for first-order optimization methods under the assumption that the weights of the network do not move much from their initialization [Jacot et al., 2018, Allen-Zhu et al., 2019, Arora et al., 2019, Chen et al., 2021, Ji and Telgarsky, 2020, Cao and Gu, 2020]. In particular, Chizat et al. [2019] recognize that "lazy training" phenomenon is due to a choice of scaling that makes the model behave as its linearization around the initialization. Interestingly, linearity has been hypothesized as key reason for existence of adversarial examples in neural networks [Goodfellow et al., 2015]. Furthermore, this was used to prove existence of adversarial attacks for random neural networks [Bubeck et al., 2021, Bartlett et al., 2021] and also provide the basis of our analysis in this work.

## 3 Main Results

In this section, we present our main result. We assume that the data is normalized so that $\|x\|_2 = 1$. We show that under the lazy regime assumption, a single gradient step on $f$ suffices to find an adversarial example to flip the prediction, given the network width is sufficiently wide but not excessively wide.

**Theorem 3.1** (Main Result). Let $w_{s,0} \sim N(0, I_d)$, $a \sim \text{unif}(\{1, +1\})$ and $\gamma \in (0, 1)$. For any given $x \in \mathbb{S}^{d-1}$, with probability at least $1 - \gamma$, the following holds simultaneously for all $W \in \mathcal{B}_{2,\infty}(W_0, C_0/\sqrt{m})$

$$\text{sign}(f(x; a, W)) \neq \text{sign}(f(x + \delta; a, W))$$

where $\delta = \eta \nabla_x f(x; a, W)$, provided that the following conditions are satisfied:

1. **Step Size:** $|\eta| = \dfrac{C_2}{\|\nabla f(x; a, W)\|^2}$,

2. **Width requirement:** $\max \left\{ d^{2.4}, C_3 \log \left( \dfrac{1}{\gamma} \right) \right\} \leq m \leq C_4 \exp(d^{0.24})$,

where $C_0, C_2, C_3, C_4$ are constants independent of width $m$ and dimension $d$.

Several remarks are in order.

We note that the setting in Theorem 3.1 is motivated by the popular gradient ascent based attack model which has been shown to be fairly effective in practice. While in practice, we perform multiple steps of projected gradient ascent to generate an adversarial perturbation, here we show that a single step of gradient suffices with a small step size of $O(1/d)$. This attests to the claim that over-parameterized neural networks trained in the lazy regime remain quite vulnerable to adversarial examples. We confirm our findings empirically in Section 5.

Our analysis suggests that an attack size of $O(1/\sqrt{d})$ suffices, i.e., for any $x$ a perturbation $\delta$ of size $\|\delta\| = O(1/\sqrt{d})$ can flip the sign of $f(x)$. This is a relatively small budget for adversarial perturbations, especially for for high-dimensional data. Indeed, in practice, we find that a noise budget of this size allows for adversarial attacks on neural networks trained on real datasets such as MNIST and CIFAR-10 [Madry et al., 2018]. Interestingly, we find in our experiments, that $O(1/\sqrt{d})$ is the "right" perturbation budget in the sense that there is a sharp drop in robust accuracy. More details are provided in Section 5.

Our investigation focuses on the settings wherein the weights of the trained neural network stay close to the initialization. Further, given the initialization scheme we consider, the $O(1/\sqrt{m})$ deviation of the incoming weight vector at any neuron of the trained network from its initialization precisely captures the neural tangent kernel (NTK) regime studied in several prior works [Ji and Telgarsky, 2020, Du et al., 2018]. This $O(1/\sqrt{m})$ deviation bound is also the largest radius that we can allow in our analysis; see Section 4 for further details. In fact, our results would also hold for adversarial training given the same initialization and the bound on the deviation of weights is met. Curiously, our empirical results exhibit a phase transition around the perturbation budget of $O(1/\sqrt{m})$ – we see in Section 5 that robust accuracy increases as we allow the network weights to deviate more.

There is an interesting interplay between the network width and the input dimensionality. Our results require that the network is sufficiently wide (i.e., polynomially large in $d$), but not excessively wide (is sub-exponential in $d$). The upper bound is large enough to allow for the over-parameterization required in the NTK setting, as long as the input dimensionality is sufficiently large.

To get an intuition about how the result follows, we first paraphrase the arguments from Bubeck et al. [2021], Bartlett et al. [2021] for random neural networks with weights $W_0$. We consider a linear model $f(w; x) = \frac{1}{\sqrt{d}} \sum_{s=1}^{d} w_{s,0} x_s$. Since the initial weights are independent (sampled i.i.d. from $\mathcal{N}(0, 1)$), using standard concentration arguments, we have that for large $d$, with high probability $|f(w_0; x)| \leq O(1)$. In contrast, $\nabla_x f(w_0; x) = w$, so $\|\nabla_x f(w_0; x)\| = \|w\|$, which is roughly $\Theta(\sqrt{d})$. Hence, for large enough $d$, a step of gradient ascent suffices to change the sign of $f$.

The above reasoning can be extended to weights that are *close* to random weights. In particular, for arbitrary weights $W$ close to the initialization, consider the linear model: $f(w; x) = \frac{1}{\sqrt{d}} \sum_{s=1}^{d} w_s x_s$.

Since $\|w_s - w_{s,0}\| = O(1)$, thus even in the worst case, the function still behaves as a constant translation of the initial prediction model $f(w_0; x)$, and thus is constant. Similarly, $\|\nabla f(w, x)\| = \Omega(\sqrt{d})$.

The work of Bubeck et al. [2021] extends the above observation to ReLU activation exploiting the fact that the function, even though non-linear, is still locally linear. Note that, in a different context, this local linearity property is central to the Neural Tangent Kernel regime. In particular, the key underlying idea there is based on the linear approximation to the prediction function. Furthermore, with appropriate initialization, a linear approximation is not only good enough model close to the initialization, but also enables tractable analysis. We collapse this "good" linearization property under the lazy training assumption since that suffices for our purposes.

**Beyond Lazy regime.** Our result can be easily extended to a broader class of neural networks. In particular, consider the cone $C = \{W : \exists r > 0, rW \in \mathcal{B}_{2,\infty}(W_0, \frac{C_0}{\sqrt{m}})\}$; see Figure 1. Note that for binary classification, scaling the network weights with a positive factor does not change the prediction. As a result, Theorem 3.1 still holds for network weights which belong to the cone $C$ but may not be in the lazy regime.

Figure 1: Cone containing the lazy-regime ball around initialization $W_0$.

**Robust Test Error.** As a corollary to Theorem 3.1, we have that none of the networks in the lazy regime are robust, i.e., for all networks in the lazy regime, the robust test error is bounded from below by a constant with high probability.

**Corollary 3.2.** Let $w_{s,0} \sim N(0, I_d)$, $a \sim \text{unif}(\{-1, +1\})$, $R = \tilde{O}(1/\sqrt{d})$ and $\gamma \in (0, 1)$. In the parameter settings of Theorem 3.1, for any data distribution $\mathcal{D}$ supported on the sphere $\mathbb{S}^{d-1}$, with probability at least $1 - 10\gamma$ (over the draw of $W_0$ and a), for all $W$ s.t. $W \in \mathcal{B}_{2,\infty}(W_0, C_0/\sqrt{m})$, the robust error $L_R(W; \mathcal{D}) \geq 0.9$.

We note that our result above is not a contradiction to the prior result of Gao et al. [2019], which claims that adversarial training finds robust classifiers in overparameterized models. Firstly, our result shows that there is no robust classifier in the lazy regime (Definition 2.1). In contrast, Gao et al. [2019] simply assume the existence of robust classifier in an NTK setting; see [Gao et al., 2019, Assumption 5.2]. Indeed, ignoring some of the (potentially unimportant) differences between our setting and that of Gao et al. [2019], (e.g., smooth vs. non-smooth activations and tied vs. non-tied weights in the last layer), then our result may suggest that the assumption in Gao et al. [2019] is incorrect. Secondly, the main result of Gao et al. [2019] is to show that the excess robust training error of the output of adversarial training is small. However, this only implies a small robust training error if the best-in-class robust training error is also small, which, again, as we discussed above, does not hold. Finally, despite the validity of the key assumption in Gao et al. [2019], the authors only provide a bound on robust training error. In contrast, our result shows that robustness may not hold in the sense of robust test error.

## 4 Proof Sketch

In this section, we provide an outline of the proof of Theorem 3.1. Recall that our goal is to show that for any given $x \in \mathbb{S}^{d-1}$, the vector $\delta = \eta \nabla_x f(x; a, W)$ is such that $\|\delta\| \ll \|x\|$, and $\text{sign}(f(x; a, W)) \neq \text{sign}(f(x + \delta; a, W))$. Assume without loss of generality that $f(x; a, W) > 0$, then $\text{sign}(\eta) = -1$. From the fundamental theorem of calculus, we have $f(x+\delta) = f(x) + \int_0^1 f'(x+t\delta)dt$. Using simple algebraic manipulations (see full proof in Appendix B), we have the following,

$$f(x+\eta\nabla f(x;a,W);a,W) \leq \overbrace{f(x;a,W)}^{O(1)} - |\eta| \overbrace{\|\nabla f(x;a,W)\|}^{\Omega(\sqrt{d})}$$

$$+ |\eta| \underbrace{\sup_{\substack{\delta \in \mathbb{R}^d: \\ \|\delta\| \leq \eta\|\nabla f(x;a,W)\|}} \overbrace{\|\nabla f(x+\delta;a,W) - \nabla f(x;a,W)\|}^{o(\sqrt{d})}}$$

The goal now is to control each of the three terms on the right hand side. Note that the suggested bounds on each of the terms will suffice to flip the label, thereby establishing the main result.

Given any x and $W \in \mathcal{B}_{2,\infty}\left(W_0, \frac{C_0}{\sqrt{m}}\right)$, we need to control:

1. The function value of neural network $f(x; a, W)$

2. The size of the gradient of neural network $\|\nabla f(x; a, W)\|$ with respect to the input x

3. The size of the difference of gradients $\|\nabla f(x; a, W) - \nabla f(x + \delta; a, W)\|$ for $\|\delta\| \leq O(1/\sqrt{d})$

## 4.1   Bounding the function value

**Lemma 4.1.** (Informal) For any x, with probability at least $1 - \gamma$, $|f(x; a, W)| \leq 2\sqrt{\log(2/\gamma)} + C_0$ holds for all $W \in \mathcal{B}_{2,\infty}\left(W_0, \frac{C_0}{\sqrt{m}}\right)$, provided that $m \geq \Omega(\log(2/\gamma))$.

**Proof Sketch:**   From triangle inequality,

$$|f(x; a, W)| \leq |f(x; a, W_0)| + |f(x; a, W) - f(x; a, W_0)|.$$

For the first term, we have from Bubeck et al. [2021] that $|f(x; a, W_0)| = O(1)$. For the second term $|f(x; a, W) - f(x; a, W_0)| \leq O(1)$ owing to the Lipschitzness of ReLU. Note that $O(1/\sqrt{m})$ is the largest deviation on $\|w_s - w_{s,0}\|$ that we can allow to get the constant bound here.

We next move to the second and third terms. We note that while bounding the first term simply follows from the Lipschitzness of the network, as we show in the following sections, the second and third terms require a finer analysis, due to the non-smoothness of ReLU.

## 4.2   Bounding the gradient

**Lemma 4.2.** (Informal) For any x, with probability at least $1 - \gamma$, $\|\nabla f(x; a, W)\| \geq \frac{1}{4}\sqrt{d}$ holds for all $W \in \mathcal{B}_{2,\infty}(W_0, \frac{C_0}{\sqrt{m}})$, provided that $m \geq \Omega(\log(1/\gamma))$ and $d \geq \Omega(\frac{\log(m/\gamma)}{m})$.

**Proof Sketch:**   Let $P := I_d - xx^\top$ be the projection matrix onto the orthogonal complement of x. Using triangle inequality,

$$\|\nabla f(x; a, W)\| \geq \|P\nabla f(x; a, W)\| \geq \|P\nabla f(x; a, W_0)\| - \|P\nabla f(x; a, W) - P\nabla f(x; a, W_0)\|.$$

The first term $\|P\nabla f(x; a, W_0)\| \geq \Omega(\sqrt{d})$, which follows from Bubeck et al. [2021] with minor changes arising from different scaling of the initialization.

We further decompose the second term,

$$\|P\nabla f(x; a, W) - P\nabla f(x; a, W_0)\| \leq \left\| \frac{1}{\sqrt{m}} \sum_{s=1}^{m} a_s P(w_s \sigma'(w_s^\top x) - w_{s,0}\sigma'(w_s^\top x)) \right\|$$

$$+ \left\| \sup_{W \in \mathcal{B}_{2,\infty}(W_0, \frac{C_0}{\sqrt{m}})} \frac{1}{\sqrt{m}} \sum_{s=1}^{m} a_s P w_{s,0}(\sigma'(w_s^\top x) - \sigma'(w_{s,0}^\top x)) \right\|.$$

The first term is $O(1)$ simply from the fact that $\|w_s - w_0\| \leq \frac{C_0}{\sqrt{m}}$. To bound the second term, the key arguments are as follows: by definition of P, the term $P w_{s,0}$ is independent of $\sigma'(w_s^\top x) - \sigma'(w_{s,0}^\top x)$, therefore, as in Bubeck et al. [2021], we can bound both of these terms independently. Note that $a_s P w_{s,0} \sim \mathcal{N}(0, I_{d-1})$. Therefore, the second term is equal in distribution to the following random variable:

$$\sup_{W \in \mathcal{B}_2(W_0, \frac{C_0}{\sqrt{m}})} \frac{\|z\|}{\sqrt{m}} \sqrt{\sum_{s=1}^{m}(\sigma'(w_s^\top x) - \sigma'(w_{s,0}^\top x))^2}$$

where $z \sim \mathcal{N}(0, I_{d-1})$.

The term $|\sigma'(w_s^\top x) - \sigma'(w_{s,0}^\top x)| = 1$ only when the $s$-th neuron has different signs for incoming weights $w_{s,0}$ and $w_s$. So, at a high-level, we want to bound the number of neurons changing sign between weights $w_0$ and $w$. To do this, we draw insights from results in the lazy training regime that

bound the number of such neurons in terms of deviations in weights. In particular, the following lemma shows that with high probability, for all $W \in \mathcal{B}_{2,\infty}(W_0, \frac{C_0}{\sqrt{m}})$, at most $O(\sqrt{m})$ neurons can have a sign different from their initialization.

**Lemma 4.3.** Let $S_v := \left\{ s \big| \exists W, W \in \mathcal{B}_{2,\infty}\left(W_0, \frac{C_0}{\sqrt{m}}\right), \mathbb{1}[\langle w_s, x \rangle > 0] \neq \mathbb{1}[\langle w_{s,0}, x \rangle > 0] \right\}$. Then, with probability at least $1 - \gamma$, we have that $|S_v| \leq C_0\sqrt{m} + \sqrt{\frac{m \log(1/\gamma)}{2}}$.

The key idea is that we can absorb the union bound in the definition of the set above without compromising any increase in the size of the set. Putting all of these pieces together yields the bound claimed in Lemma 4.2.

### 4.3 Bounding the difference of gradients

**Lemma 4.4.** For any x, with probability at least $1 - \gamma$, for all $W \in \mathcal{B}_{2,\infty}\left(W_0, \frac{C_0}{\sqrt{m}}\right)$, we have that $\sup_{r \in \mathbb{S}^{d-1}, \delta \in \mathbb{R}^d} \|\nabla f(x; a, W) - \nabla f(x + \delta; a, W)\| \leq o(\sqrt{d})$, provided that $\|\delta\| \leq O(1/\sqrt{d})$, and $d^{2.4} \leq m \leq O(\exp(d^{0.24}))$.

**Proof Sketch:** It follows from the definition of norm that

$$\|\nabla f(x; a, W) - \nabla f(x + \delta; a, W)\| = \sup_{r \in \mathbb{S}^{d-1}, \delta \in \mathbb{R}^d} \frac{1}{\sqrt{m}} \sum_{s=1}^m a_s w_s^\top r \left( \sigma'(w_s^\top x) - \sigma'(w_s^\top(x + \delta)) \right).$$

The strategy, then, is to bound the right-hand side for fixed $r$ and $\delta$, and then use an $\varepsilon$-net argument to bound the supremum. To that end, define the following quantities:

$$X_{s,0} = a_s w_{s,0}^\top r(\sigma'(w_{s,0}^\top x) - \sigma'(w_{s,0}^\top(x + \delta))), X_s = a_s w_s^\top r(\sigma'(w_s^\top x) - \sigma'(w_s^\top(x + \delta))).$$

Then, the norm of the difference of the gradients is upper bounded by

$$\frac{1}{\sqrt{m}} \sum_{s=1}^m X_{s,0} + \frac{1}{\sqrt{m}} \sum_{s=1}^m (X_s - X_{s,0}).$$

The first term $\frac{1}{\sqrt{m}} \sum_{s=1}^m X_{s,0} = O((R\sqrt{\log(d)})^{1/4})$ which follows from Bubeck et al. [2021] with minor changes due to the different scale of the initialization. However, the second term requires a more careful analysis. In particular, the difference $X_s - X_{s,0}$ includes deviations of both variables x and W. Therefore, we need to expand the difference in terms of each variable and carefully bound the resulting terms. Similar to the ideas presented in the previous section, here we need to bound the size of the set of neurons that can change sign under perturbations on both x and W. It turns out that the size of this set is in the same order as the one in Lemma 4.3, as long as the perturbation in x is $O(1)$.

## 5 Experiments

The primary goal of this section is to provide empirical support for our theoretical findings. In particular, we want to verify how tight our theoretical bounds are in practice, in terms of the size of the perturbation needed and the interplay of robustness with training in the lazy regime. For instance, are the trained models indeed vulnerable to small perturbations of size $O(1/\sqrt{d})$ as suggested in Theorem 3.1? Is the assumption of training in the lazy regime merely for analytical convenience or do we see a markedly different behavior if we allow the iterates to deviate more than $O(1/\sqrt{m})$ from initialization?

**Datasets & Model specification.** We utilize the MNIST dataset for our empirical study. MNIST is a dataset of $28 \times 28$ greyscale handwritten digits [LeCun et al., 1998]. We extract examples corresponding to images of the digits '0' and '1', resulting in 12665 training examples and 2115 test examples. To study the behaviour of different quantities as a function of the input feature dimensionality, we downsample the images by different factors to simulate data with $d$ ranging between 25 and 784 (the original dimension). We use two-layer ReLU networks with a fixed top layer to match our theory and use the initialization described in Section 2.

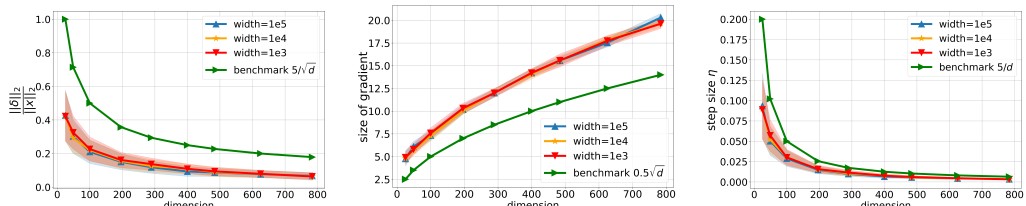

Figure 2: Minimal size of the perturbation vector needed to flip the predicted label (**left**), the norm of the gradient $\nabla f(\mathrm{x}; \mathrm{a}, \mathrm{W})$ for W trained in the lazy regime (**middle**), and the corresponding step size (**right**), as a function of the input dimension $d$ for a fixed value of $C_0 = 10$ and different values of network widths ($m$).

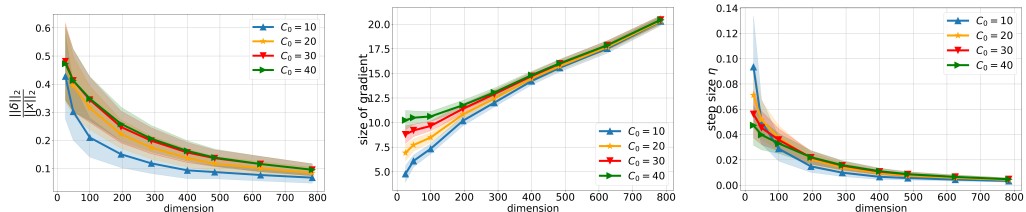

Figure 3: Minimal size of the perturbation vector needed to flip the predicted label (**left**), the norm of the gradient $\nabla f(\mathrm{x}; \mathrm{a}, \mathrm{W})$ for W trained in the lazy regime (**middle**), and the corresponding step size (**right**), as a function of the input dimension $d$ for a fixed value of network width of $m = 10^5$ and different values of $C_0$.

## 5.1 Lazy regime for standard SGD

**Setting.** For each $d \in \{5^2, 7^2, 10^2, 14^2, 20^2, 25^2, 28^2\}$, we set the width of the network $m \in \{1e3, 1e4, 1e5\}$. We train the network using standard SGD on the logistic loss using a learning rate of $0.1$. We denote the weight matrix at the $t^{\text{th}}$ iterate of SGD as $\mathrm{W}_t$ and the incoming weight vector into the $s^{\text{th}}$ hidden node at iteration $t$ as $\mathrm{w}_{s,t}$. The training terminates if the iterates exit the lazy training regime, i.e., if $\sup_{1 \le s \le m} \|\mathrm{w}_{s,t} - \mathrm{w}_{s,0}\| > \frac{C_0}{\sqrt{m}}$. We experiment with $C_0 \in \{10, 20, 30, 40\}$. We note that regardless of the above stopping criterion, each of the trained models achieves a test error of less than $1\%$. In other words, the models that our training algorithm returns are guaranteed to satisfy the lazy regime assumption and generalize well.

Given a trained neural network with parameters $(\mathrm{W}, \mathrm{a})$, for each test example $\mathrm{x}$, we find the smallest step size $\eta$ such that the perturbation $\delta = \eta \nabla_{\mathrm{W}} f(\mathrm{x})$ flips the sign of the prediction, i.e., $f_{\mathrm{W}}(\mathrm{x}) f_{\mathrm{W}}(\mathrm{x} + \delta) < 0$. For each experimental setting (i.e, for each value of $d$, $C_0$, and $m$), we report results averaged over 5 independent random runs.

**Results.** The key quantities in our analysis are the step size $\eta$, the norm of the gradient $\|\nabla f(\mathrm{x}; \mathrm{a}, \mathrm{W})\|$, and the size of the perturbation vector which is given as the product of the step size and the norm of the gradient. In Figure 2, we plot these quantities as a function of input dimension $d$, for a fixed $C_0$, for various values of width $m \in \{1e3, 1e4, 1e5\}$ (corresponding to the curves in red, orange, and blue, respectively). The green curves represent the corresponding theoretical bound. As predicted by Theorem 3.1, the gradient norm behaves roughly as $\Omega(\sqrt{d})$. Furthermore, the minimum step-size required to flip the label, and the minimum size of the perturbation vector (normalized by the input norm) behave as $O(1/d)$ and $O(1/\sqrt{d})$, respectively.

In Figure 3, we explore the behaviour of the key quantities as we allow for larger deviations in the lazy regime, by allowing for larger values of $C_0$, for a fixed network width of $m = 10^5$. We see that as $C_0$ increases, both the size of the gradient and the minimum step size required to flip the label increase, thereby increasing the size of the perturbation vector needed to change the sign of the prediction. This suggests that allowing the trained networks to deviate more from the initialization potentially leads to models that are more robust (against the attack model based on a single step of gradient ascent).

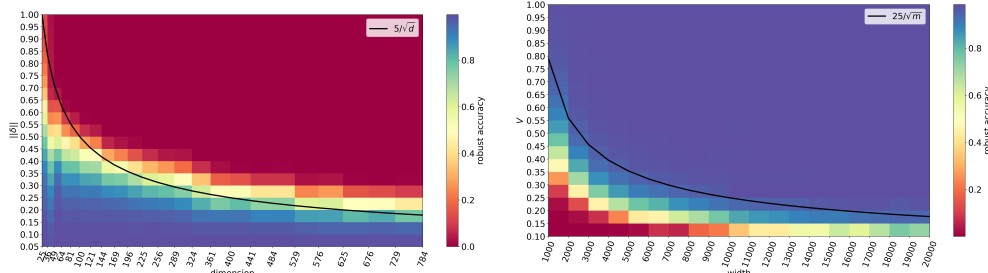

Figure 4: **(Left):** Robust test accuracy as a function of perturbation size $\|\delta\|$ and dimension $d$, for a fixed network width of $m = 1000$ and a fixed value of maximal deviation of weight vectors of $V = \frac{C_0}{\sqrt{m}}$ with $C_0 = 10$. **(Right):** Robust test accuracy as a function of width $m$ and maximal deviation of weight vectors, $V$, for a fixed dimension of $d = 784$ and fixed perturbation size $\|\delta\| = 0.2$.

Remarkably, the minimum perturbation size, closely follows the $O(1/\sqrt{d})$ behaviour predicted by the theorem. This begets further investigation into the required perturbation size, which is the focus of the next subsection.

## 5.2 Lazy regime for adversarial training

We begin by noting that our theoretical results are not limited to any particular learning algorithm. In fact, the results hold simultaneously for all networks in the lazy regime. In this section, we further verify our theoretical results for networks that are adversarially trained in the lazy regime.

**Setting.** We downsample images as in the previous section to simulate data with $d \in \{5^2, 6^2, 7^2, \ldots, 28^2\}$. For this set of experiments, we also normalize the data, so that $\|\mathbf{x}\| = 1$, to closely match the setting of our theorem. We use projected gradient descent (PGD) [Madry et al., 2018] for adversarial training. To ensure that the updates remain in the lazy regime, after each gradient step, we project the weights onto $\mathcal{B}_{2,\infty}(W_0, V)$; see Algo. 1 in the appendix for details. We experiment with values of $V$ and the size of $\|\delta\|$ in the intervals $0.05 \leq V \leq 1$ and $0.05 \leq \|\delta\| \leq 1$, respectively. These ranges are wide enough to include the $O(1/\sqrt{d})$ and $O(1/\sqrt{m})$ theoretical bounds that we aim to verify here. We note that the projection step is not commonly used in adversarial training; however, in our experiments, it does not hurt the test accuracy of the trained models, perhaps due to the simplicity of the classification task. At test time, we generate the adversarial examples using the PGD attack instead of a single gradient attack as in Section 5.1. We report the robust test accuracy averaged over 5 independent random runs of the experiment.

**Results.** For the experiments reported in Figure 4, left panel, we fix the network width to $m = 1000$ and set $V = \frac{C_0}{\sqrt{m}} = \frac{10}{\sqrt{1000}} = 0.316$, to check how the dimension $d$ and the perturbation budget affect the robust accuracy. We observe a sharp drop in robust accuracy about the $5/\sqrt{d}$ threshold for the perturbation budget $\|\delta\|$ as predicted by Theorem 3.1. We can see that for perturbation sizes slightly larger than $5/\sqrt{d}$, the robust accuracy drops to 0, while the trained networks are robust against smaller perturbations.

For the experiments reported in the right panel of Figure 4, we fix the input dimensionality to $d = 28^2$, and the perturbation size to $\|\delta\| = 0.2$, to check how the network width $m$ and the maximal deviation in weights, given by $V$, affect the robust test accuracy. We observe a phase transition in the robust test accuracy at value of $V$ around $O(1/\sqrt{m})$, as required by Theorem 3.1. In particular, for any fixed width $m$, even a slight deviation from the $25/\sqrt{m}$ radius allows for training networks that are completely robust against the PGD attacks. With a smaller radius, however, the robust test accuracy quickly drops to 0. This experiment suggests that adversarial robustness may require going beyond the lazy regime.

**Why MNIST?** We remark that our empirical demonstration on a simple dataset makes a stronger case for our theoretical result. This is because the perturbation size of $O(1/\sqrt{d})$ prescribed by our

theoretical result is **worst-case**, regardless of the dataset. So, in an easy setting like binary-MNIST, we would expect our bound to be pessimistic and a smaller perturbation to be sufficient. On the contrary, we see that even MNIST exhibits a perturbation-dimension relationship that closely follows our theory. This can be regarded as an empirical evidence of the tightness of our bounds.

## 6    Conclusion

In this paper, we study the robustness of two-layer neural networks trained in the lazy regime against inference-time attacks. In particular, we show that the class of networks for which the weights are close to the initialization after training, are still vulnerable to adversarial attacks based on a single step of gradient ascent with a perturbation of size $O(\frac{\|\mathbf{x}\|}{\sqrt{d}})$.

Our works suggests several research directions for future work. First, the focus here is on two-layer networks, so a natural question is to ask whether the same holds for multi-layer networks. Second, while we show that a single step of gradient ascent-based attack is sufficient for our purposes, we expect to get stronger results if we consider stronger attacks, for instance, with gradient ascent based attack that is run to convergence. Third, our experiments highlight an intriguing relationship between the width, the input dimension, and the robust accuracy. So, a natural avenue is to understand these dependencies more carefully. These include, figuring out the dependence on the input dimensionality of the minimal perturbation required to generate an adversarial example. Similarly, how does the robust error behave as a function of the maximal distance of the weight vectors from the initialization.

## Acknowledgments and Disclosure of Funding

This research was supported, in part, by the DARPA GARD award HR00112020004, NSF CAREER award IIS-1943251, and the Spring 2022 workshop on "Learning and Games" at the Simons Institute for the Theory of Computing.

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
