# OpenReview forum: "Adversarial Robustness is at Odds with Lazy Training"
_NeurIPS.cc/2022/Conference — NeurIPS 2022 Accept_

### Official Review · Reviewer_TJvx · 2022-07-10

**Rating:** 5
**Confidence:** 3
**Soundness:** 3 good
**Presentation:** 3 good
**Contribution:** 3 good

**Summary:**

The paper takes a solid step towards explaining adversarial sensitivity of neural networks based on random networks. The paper shows that networks can be attacked by a single gradient descent step in the lazy regime where parameters are close to initialization.

**Questions:**

See the weakness part.

**Limitations:**

The authors adequately addressed the potential negative societal impact of their work? I think fixing $a$ is a big limitation for the technical results.

**Strengths And Weaknesses:**

Strengths

1. The paper takes a solid step towards explaining adversarial sensitivity of neural networks based on random networks [1]. In the lazy regime like NTK, while networks can show good clean perfomrance, they can easily attacked by a single gradient step.

2. The paper is easy to follow with clear presentation.

3. I appreciate the adversarial training in the empirical experiments, which shows that the theorem holds for arbitrary $w$ near initialization not constraint by the training algorithms.

Weaknesses

1. My major concern is that the Theorem only optimizes the first layer $W$ while $a$ is not included. In the experiment, $a$ is also a tunable parameter. This is inconsistent. Besides, allowing tunable $a$ may bring contradictory result. In [2], the authors show that adversarial training is possible to converge in the overparameterized models. It seems to indicate that adversarial robustness is not impossible in the lazy regime.

2. I recommend the authors to report clean performance in the experiment parts to show the network actually achieves good clean generlization near initialization.

3. One conclusion is misaligned with empirical findings. The adversarial trained networks need larger size of adversarial example to attack the network (Figure 4(b) in [3]). And the common $L_2$ adversarial budget set for ImageNet (dimension is $224\times224\times 3$) is also higher than CIFAR10 (dimension is $32\times32\times 3$). This is contradictory to the conclusion that the label will be flipped with $O(\frac{1}{\sqrt{d})$ adversarial noise.

[1] S´ebastien Bubeck et.al. A single gradient step ﬁnds adversarial examples on random two-layers neural networks, NeurIPS 2021

[2] Ruiqi Gao et.al. Convergence of Adversarial Training in Overparametrized Neural Networks, NeurIPS 2019

[3] Ali Shafahi et.al. Are adversarial examples inevitable?, ICLR 2019

---

> ### Author Response · Authors · 2022-08-02
> **Answer to weakness**
>
> 1.  Actually, what you say is not correct. In the experiment, the top layer a is also fixed through the training, which is consistent with the theorem. See lines 275-276. Our paper does not contradict the result of [2]. We consider the lazy regime as $\|W-W_0\|_{2,\infty}\leq \frac{R}{\sqrt{m}}$, whereas [2] considers the setting $\|W-W_0\|_F\leq R$. Hence, note that the set of allowed models in [2] is larger than ours and therefore the networks found may not lie in the lazy regime as per our definition.
>
> 2. We do state in the paper that each of the trained models achieves a test error of less than 1%. See lines 283-284.
>
> 3. Note that our result that $O(1/\sqrt{d})$ perturbation size suffices holds for networks trained in the lazy regime. This may or may not be true in general. It is likely that networks in prior works do not fall within the lazy regime. Hence, there is no contradiction if they do not follow the $O(1/\sqrt{d})$ perturbation size.

---

> > ### Comment · Reviewer_TJvx · 2022-08-06
> > **Response to Authors**
> >
> > 1. Thanks for clarifying the experiment setting and results. I agree with the authors that the experiment setting is the same as the theory and trained models achieves good clean test error.
> > 2. I would suggest the authors to limit their claim in the abstract and title. As the lazy regime described in [2] is also reasonable, in my humble opinion, the claims that "Adversarial Robustness is at Odds with Lazy Training" may not be true for all lazy regimes. The argument in the title and abstract may be too strong.
> > 3. My third weakness is that the paper may not give insights beyond lazy regimes for the empircal studies. After reading authors' response, I still think it is a weakness.
> >
> > Overall, I think it is a technically strong paper. I am open for discussion about the relationship between this paper and [2]. I will increase my score if the authors are able to limit their claims about the lazy regime or convince me that [2] does not indicate robustness can be achieved in the lazy regime.

---

> > > ### Author Response · Authors · 2022-08-08
> > > **Response on new comments**
> > >
> > > We thank the reviewer for their response to our comments, and appreciation of our efforts.
> > >
> > > For the simplicity of the discussion, we denote **Setup-1** as $\|W-W_0\|_{2,\infty}\leq \frac{R}{\sqrt{m}}$ (lazy regime in our paper), and **Setup-2** as $\|W-W_0\|_F\leq R$ (lazy regime in [2]).
> > >
> > > The following are the key differences between our result and that of [2]:
> > >
> > > 1. Existence of robust classifier: Our result shows that there is no robust classifier in **Setup-1** regardless of the training dataset. In contrast, [2] simply assumes that this is true in **Setup-2** –  see Assumption 5.2 in [2]. If we ignore the (potentially un-important) difference in our settings – smooth/non-smooth activation and tied/non-tied weights in the last layer – then our result shows that this assumption is incorrect. In fact, the prior work of [1] on the existence of adversarial examples in random neural networks suffices to establish the incorrectness of this assumption since their assumption is basically about networks at initialization. We feel that this observation actually makes our result stronger, and we will add a discussion to this effect in the revised version.
> > >
> > > 2. Training error vs excess training error: The main result of [2] is to show that the excess robust training error of the output of adversarial training is small; however, this only implies a small robust training error if the best in-class robust training error is also small. As discussed above, [2] simply assumes it to be true however ours (and prior work) show that it is not necessarily the case. Therefore, a bound on excess robust training error may not be interesting.
> > >
> > > 3. Training error vs test error: Furthermore, even with their (potentially false) assumption, [2] only gives a bound on robust training error whereas our result shows that robustness may not hold even in the sense of robust test error.
> > >
> > > 4. Finally, we argue that our definition of lazy regime (Def 2.1), i.e., the **Setup-1** above is common in the NTK literature. We cite here several prominent works: [3, Theorem 2], [4, Lemma 5.3], [5, Lemma 4.2]. There are some works that also define the lazy regime w.r.t. Frobenius norm (for instance,  [6, 7]) to prove generalization guarantees for SGD – however, they still consider a much smaller set than **Setup-2** ). In particular, they assume that $\|W-W_0\|_F\leq \frac{R}{\sqrt{m}}$ (see [6, Lemma 4.3], [7, Corollary 4.9]). As a result, it is not even entirely clear whether the **Setup-2** gives you a small clean generalization error, whereas our **Setup-1** is the exact same setting as [3], which guarantees a small clean generalization error.
> > >
> > > Regardless, if you still feel strongly we will be happy to tone down the claim and/or add a discussion comparing with the literature [2]. Let us know if you have a preference.
> > >
> > >
> > > [1] Bubeck, Sébastien, et al. "A single gradient step finds adversarial examples on random two-layers neural networks." Advances in Neural Information Processing Systems 34 (2021): 10081-10091.
> > >
> > > [2] Gao, Ruiqi, et al. "Convergence of adversarial training in overparametrized neural networks." Advances in Neural Information Processing Systems 32 (2019).
> > >
> > > [3] Ji, Ziwei, and Matus Telgarsky. "Polylogarithmic width suffices for gradient descent to achieve arbitrarily small test error with shallow relu networks." arXiv preprint arXiv:1909.12292 (2019).
> > >
> > > [4] Arora, Sanjeev, et al. "Fine-grained analysis of optimization and generalization for overparameterized two-layer neural networks." International Conference on Machine Learning. PMLR, 2019.
> > >
> > > [5] Zou, Difan, and Quanquan Gu. "An improved analysis of training over-parameterized deep neural networks." Advances in neural information processing systems 32 (2019).
> > >
> > > [6] Cao, Yuan, and Quanquan Gu. "Generalization bounds of stochastic gradient descent for wide and deep neural networks." Advances in neural information processing systems 32 (2019).
> > >
> > > [7] Cao, Yuan, and Quanquan Gu. "Generalization error bounds of gradient descent for learning over-parameterized deep relu networks." Proceedings of the AAAI Conference on Artificial Intelligence. Vol. 34. No. 04. 2020.

---

> > > > ### Comment · Reviewer_TJvx · 2022-08-08
> > > > **Response to Authors**
> > > >
> > > > I appreciate the responses from the authors. It convinces me that [2] does not indicate robustness can be achieved in the lazy regime. I will increase my score accordingly.

---

> > > > > ### Author Response · Authors · 2022-08-08
> > > > > **Response to reviewer**
> > > > >
> > > > > Thank you for your quick response and for engaging in the discussion. Given that we have addressed all your concerns and that you can appreciate now why this result is important given that our work also sheds light on works like [2], we were hoping that you would be more supportive of our work. Your rating of 5 seems harsh in light of all the clarifications. We hope you will reconsider it. Thanks!

---

### Official Review · Reviewer_7pem · 2022-07-10

**Rating:** 5
**Confidence:** 4
**Soundness:** 3 good
**Presentation:** 2 fair
**Contribution:** 3 good

**Summary:**

This paper studies model robustness in the context of lazy training, i.e, the model parameters are in the neighborhood of its initialization.
Specifically, the authors find that the lazy trained models are quite vulnerable against adversarial attacks, although it has good generalization on clean inputs.
It is proved that a single step attack with a small step size can flip the prediction of the two-layer networks under lazy training. This claim is validated in the numerical stimulation.

**Questions:**

I have pointed out the major questions in the weakness part in the previous section. In addition, I have one additional question:

1. In the right part of Figure 2, why the step size $\eta$ is significantly bigger in the small $C_0$ and small $d$ cases? This is not consistent with what the authors say.

In addition, the presentation of this paper needs improvements, such as:

1. Definition of $L_R$ (between line 95 and 96), $L_R$ is not a function of $(x, y)$ because they are averaged for expectation calculation.

2. Theorem 3.1, $C_1$ is not defined.

3. Corollary 3.2, line 176. it should be $a \sim unif\\{-1, +1\\}$

4. Lemma 4.4,  line 251, $c$ is not defined.

**Limitations:**

The authors briefly discussed the limitation of this work in the end of the paper.

**Strengths And Weaknesses:**

Strengths:

1. It is interesting to study the model's robustness in the context of lazy training, which makes the model efficiently learnable.
2. The theoretical proof is sound.

Weakness:

I have several concerns listed below, addressing them can improve the manuscript.

1. In Theorem 3.1, the width needs to be both upper bounded and lower bounded (Line 145). However, the lower bounds increases with the decrease of $\gamma$, and the upper bound decreases with the decreases of $\gamma$. That is to say, in some cases, the $\gamma$ can not be arbitrarily small, i.e. $\gamma \to 0$ may make $m$ infeasible under the constraints of this theorem. This is a critical concern especially in the case of small input dimension $d$. The author should comprehensively discuss this case.

2. The main theorem (Theorem 3.1) of this paper is applicable to any model parameters in the neighborhood of the initialization. However, in the empirical experiments only study the cases of training or adversarial training, could it be the vulnerability as shown in the experiments arise from the training methods, i.e, the parameters prior encoded by your training objective and optimizer?

3. In general, I think the real interesting part of this work is to show there is no robust models in the neighborhood of a random Gaussian initialization, **even with adversarial training**. I suggest the author highlight the adversarial training part, because in vanilla training, even without constraining the model parameters in the initial neighborhood, the model is not robust at all.

4. For the presentation of Theorem 3, the ``constant'' $C_2$, $C_3$, $C_4$, and $C_5$ depends on $C_0$. I agree that for notation simplicity, there is no need to provide the exact analytical form of these values. However, I do suggest the authors discuss how these values increases or decreases with $C_0$. In addition $C_1$ is not defined in this Theorem.

In general, I like the topic and the problem settings of this paper, but there are several concerns pointed in this section and the next one. It can be a good paper if the concerns above are adequately addressed. I am happy to discuss with the authors in the rebuttal and re-evaluate this paper after that.

---

> ### Author Response · Authors · 2022-08-02
> **Answer to weakness and questions**
>
> **Answer to weakness.**
>
> 1. Thank you for your careful reading. The $\gamma$ term in the upper bound is a typo; we do not need that upper bound. Thus the problem you point out no longer exists.
>
> 2. As per our theoretical result, as long as the method produces a network in the lazy regime, we expect it to be vulnerable to our attack. This result does not depend on a choice of a training algorithm. We tried SGD and adversarial training since these are two common methods for training neural networks. If the reviewer has other methods in mind , we would be happy to try them too.
>
> 3. The reviewer is correct, and we indeed mention the adversarial training result on lines 166-167. Our result focused on lazy regime, which is oblivious to the training method of neural networks, and we mention it on line 309.
>
> 4. We treat $C_0$ as a constant because we treat width and dimension as problem parameters and our focus is to characterize the result in terms of these, width and dimension. However, it is indeed possible to identify the relationship between $C_0$ and other parameters.
>
> **Answer to questions.**
>
> Firstly, the dependence of step size on dimension is consistent with our theorem. Note that our theorem says the step size of O(1/d) suffices to flip the label. So as the dimension increases, the step size decreases. In order to interpret the dependence of $C_0$, look at the left figure in figure 2, which plots the size of the perturbation. Intuitively, as the neural network goes away from random initialization (larger $C_0$), it requires a larger perturbation size to flip the sign. This behaviour is captured in this plot.
>
> We will fix these typos as suggested.

---

> > ### Author Response · Authors · 2022-08-08
> > **Open to discussion**
> >
> > Please let us know whether we address your concerns. We are happy to have more discussions for your future questions. We hope that you might consider raising your score, since you seem to appreciate our novelty and contributions.

---

> > ### Comment · Reviewer_7pem · 2022-08-08
> > **Response to the authors**
> >
> > I thank the authors for their responses and clarification. The answers address most of my concerns.
> >
> > In addition, I would suggest the authors provide more qualitative interpretations of the bounds in Theorem 1, which I believe will greatly help the readers understand the key idea of this paper.
> >
> > Overall, I think this is an interesting paper, so I increase my score from 4 to 5.

---

### Official Review · Reviewer_ztAA · 2022-07-11

**Rating:** 6
**Confidence:** 3
**Soundness:** 3 good
**Presentation:** 3 good
**Contribution:** 2 fair

**Summary:**

This paper studies the robustness of two-layer NNs against adversarial examples in the “lazy-training” regime, extending a line of work which studied this for randomly initialized two-layer NNs. The paper shows that if after training, the weights of the NN are close to the randomly initialized weights, single-step adversarial perturbations of size O(1/\sqrt(d)) flips the prediction of this NN. The authors empirically validate their theory via experiments on MNIST.


**Questions:**

- Would similar results hold for multi-class classification problems?
- It would be more convincing to see empirical evidence supporting the theory on datasets beyond MNIST (e.g., CIFAR-10 or ImageNet), and in general on tasks beyond image classification. Is there a reason why the authors didn’t try these?


**Limitations:**

The  authors adequately addressed the limitations of their work. They didn’t include potential negative societal impact though. While I don’t think there is a direct negative societal impact of this paper, I believe the authors might be able to talk about how readers might get a wrong perspective of non-robustness of NNs in general. The readers should be aware that the claims of this paper hold only for two layer NNs and in the lazy training regime only, i.e., that there is no proof yet that it is guaranteed to find adversarial examples for general NNs.


**Strengths And Weaknesses:**


#### Strengths:
- The findings in the paper are original to the best of my knowledge. The paper extends a line of work that theoretically shows that NNs are susceptible to single-step adversarial attacks.
- The claims of the paper are sounds and are well-supported by the experiments.
- I think the work is decently significant as it is the first paper to attempt to bring the theory of robustness of neural networks beyond the random-initialization regime.
- The paper is very well written and flows smoothly. I didn’t check the proof sketch in detail, but everything else was easy to grasp and follow.
- The paper lists a number of potential research directions for building on top of their work which I think is helpful and puts the contributions of the paper into context.

#### Weaknesses:
- Overall the paper is pretty good, but it can benefit from more experiments to support the theory (more datasets, multiple tasks beyond image classification)
- A discussion of extension to multi-class classification problems is missing.


#### Minor changes:
- Line 73: repeated “a”.
- Equation after line 93: extra parentheses next to the zero
- Line 134-158-164: references should be inside parentheses
- Line 146: C1 doesn’t show up in theorem 3.1 nor in the main paper.
- Line 141 and 176: a ∼ unif{1, +1} -> a ∼ unif({-1, +1})

---

> ### Author Response · Authors · 2022-08-02
> **Answer to weakness and questions**
>
> Currently, our lazy regime definition only suffices for two-layer neural networks. Complicated datasets such as Cifar10 and Imagenet require convolutional neural networks to even achieve a good clean test accuracy, and it’s currently unclear how to adapt the lazy regime definition to convolutional neural networks. The whole reason for using MNIST is to verify our theorem and we give reasons for this choice on lines 336-341. Adapting our results to multi-class classification and beyond image classification therefore requires more work and is currently out of the scope.

---

> > ### Comment · Reviewer_ztAA · 2022-08-09
> > **Thanks for clarification**
> >
> > I thank the authors for clarifying my concerns. I will keep my score the same.

---

### Official Review · Reviewer_e718 · 2022-07-12

**Rating:** 5
**Confidence:** 2
**Soundness:** 3 good
**Presentation:** 3 good
**Contribution:** 2 fair

**Summary:**

This paper studies the existence of adversarial examples (found via a single gradient step) for two-layer ReLU networks in settings where the trained weights remain close to their initialized values. The authors prove that adversarial examples exist in this specific case, and support their theoretical findings via experiments on MNIST.


**Questions:**

- How does this perturbation budget relate to what is typically considered an indistinguishable attack in adversarial training studies on MNIST?

Minor suggestions
- Typo line 26: lead -> led, incomplete sentence line 47, extra “a” in line 73, missing \citep throughout
- All the plots in Figures 1-3 are difficult to read -- they should be enlarged.


**Limitations:**

Yes

**Strengths And Weaknesses:**

Strengths
- The paper is mostly well written.
- This paper presents a novel theoretical analysis of adversarial robustness in the context of what is referred to as the "lazy regime" in this work, where the trained weights do not stray far from their initialization.
- The authors present empirical results on MNIST that back up their theoretical findings.

Weaknesses
- It's not immediately clear what the importance/significance of this work is, and how this extends beyond two-layer ReLU networks, and this very specific setting.
- The proof sketch reads a bit rushed -- the readability could be improved here.

---

> ### Author Response · Authors · 2022-08-02
> **Answer to weakness and questions**
>
> **Answer to weakness:**
>
> We thank the reviewer for their feedback. Regarding the significance of our work, we reiterate that a recent line of work has focused on finding adversarial examples for 2 layer random ReLU neworks.  Our work takes the first step towards pushing those results to machine learning settings and explains why neural networks trained by SGD that are guaranteed to generalize well can still be vulnerable to adversarial attacks.
>
> Regarding extension beyond two-layer ReLU networks, we note both elements of our work,  (a) Neural Tangent Kernel (NTK) setting and (b) adversarial examples on randomly initialized networks, have been extended to multi-layer settings (see [1,2,3] for references), We therefore expect that it should be possible to also extend our results two-layer ReLU networks, which we list it as one future direction, see line 347-348.
>
> [1] Cao, Yuan, and Quanquan Gu. "Generalization error bounds of gradient descent for learning over-parameterized deep relu networks." Proceedings of the AAAI Conference on Artificial Intelligence. Vol. 34. No. 04. 2020.
>
> [2] Zou, Difan, et al. "Gradient descent optimizes over-parameterized deep ReLU networks." Machine learning 109.3 (2020): 467-492.
>
> [3] Bartlett, Peter, Sébastien Bubeck, and Yeshwanth Cherapanamjeri. "Adversarial examples in multi-layer random relu networks." Advances in Neural Information Processing Systems 34 (2021): 9241-9252.
>
>
> **Answer to questions:**
>
> Our theory and experiments suggest that a corruption of size $O(1/\sqrt{d})$ is sufficient. In high $d$, this would be considered imperceptible – the prior work on adversarial examples for random ReLU networks also takes the same view. We also note that prior works on generating L2 perturbation attacks on MNIST use perturbation of size roughly 1.5 to achieve small robust accuracy. In contrast, in our experiments, we use a smaller L2 perturbation of size 0.2.
>
> However, we’d like to emphasize that it is not entirely fair to compare our perturbation and those in adversarial training studies on MNIST, because the neural networks used in such studies likely do not fall within the lazy regime as we discussed in this paper. Hence, it isn’t surprising they need a larger perturbation size to misclassify the neural network.
>
>
> [1] Schott, Lukas, et al. "Towards the first adversarially robust neural network model on MNIST." arXiv preprint arXiv:1805.09190 (2018).
>
> **Minor Suggestions:**
> Thanks for pointing out the typos. We will make these corrections.

---

> > ### Author Response · Authors · 2022-08-08
> > **Open to discussion**
> >
> > Please let us know whether we address your concerns. We are happy to have more discussions for your future questions. We hope that you might consider raising your score, since you seem to appreciate our novelty and contributions.

---

### Author Response · Authors · 2022-08-06
**Open to discussion**

We thank all reviewers for their careful reading of the paper and their detailed feedback. We note that all reviewers appreciated the novelty of the setting and acknowledged our contributions. We hope we have addressed all concerns and will stay available to clarify if you have more questions.

---

### Meta-Review · Area_Chair_z9oB · 2022-08-26

**Recommendation:** Accept
**Confidence:** Certain

**Metareview:**

While the overall scores are somewhat borderline, there seem to be no major flaws and all reviewers recommend accepting the paper. Hence I will follow this recommendation.

**Award:**

No

---

### Decision · Program_Chairs · 2022-09-14

Accept